# Understanding Barriers to Hepatitis C Antiviral Treatment in Low–Middle-Income Countries

**DOI:** 10.3390/healthcare13010043

**Published:** 2024-12-30

**Authors:** Rashmi Venkatesh, Andrew S. Huang, Kiya Gurmessa, Edbert B. Hsu

**Affiliations:** 1College of Osteopathic Medicine, William Carey University, Hattiesburg, MS 39401, USA; rvenkatesh548540@student.wmcarey.edu; 2Center for Global Emergency Care, Johns Hopkins University, Baltimore, MD 21209, USA; ahuang55@jh.edu; 3Bloomberg School of Public Health, Johns Hopkins University, Baltimore, MD 21205, USA; kgurmes1@alumni.jh.edu; 4School of Medicine, Johns Hopkins University, Baltimore, MD 21209, USA

**Keywords:** hepatitis C, low–middle-income countries, health barriers, knowledge gaps

## Abstract

**Background**: Direct-acting antiviral agents (DAAs) have significantly reduced Hepatitis C Virus (HCV) transmission and improved health outcomes since their FDA approval in 2011. Despite these advances, over 70 million people remain untreated globally, with a disproportionately high burden in low- and middle-income countries (LMICs). **Methods**: Through a structured search of open access informational sources and an informal peer-reviewed literature review, HCV treatment barriers were identified, compiled, and analyzed. Current challenges to HCV treatment were organized by themes and summarized as recommendations for LMICs. **Results**: Key obstacles to HCV treatment in LMICs are identified, with the underdiagnosis and undertreatment of the disease linked to inadequate funding and healthcare infrastructure for screening and testing, poor awareness among healthcare providers, and the misinformation and stigmatization of HCV disease. **Discussion**: Recommendations for LMICs to attenuate treatment obstacles include distributing educational media, implementing mobile clinics, and fostering international partnerships. The successful implementation of these interventions has been demonstrated in developed countries. **Conclusions**: To achieve the WHO’s goal of eliminating HCV as a public health threat by 2030, concerted efforts are needed by LMICs to reduce gaps in care and ensure that all patients are afforded access to testing and treatment.

## 1. Introduction

Recent advances in treatment, healthcare delivery, and other services have generally led to steady improvement in health outcomes related to the Hepatitis C Virus (HCV) in developed countries. However, currently, HCV remains a global health problem. An infectious disease primarily transmitted through infected blood with seven different genotypes and nearly one hundred different subtypes, HCV is the main etiology of chronic hepatitis. HCV can compromise overall liver function, leading to cirrhosis, hepatocellular carcinomas, and eventually death [1]. Coordinated efforts in policy and logistics have facilitated greater access to care, but no single treatment has been more instrumental in reducing the transmission of HCV and improving overall health outcomes than direct-acting antiviral agents (DAAs). Since their introduction and approval by the United States Food and Drug Administration (FDA) in 2011, the past decade has seen not only the further development of DAAs as a viable treatment option but also the increased accessibility and affordability of these drugs. This has contributed significantly to the decline in HCV prevalence, particularly in developed countries. Upon breakthroughs with second-generation DAAs, including sofosbuvir in 2013, DAA utilization expanded to more low- and middle-income countries (LMICs) [2]. By 2015, nations such as Pakistan and Georgia saw HCV treatment rates double and triple, respectively [2]. In 2016, DAA treatment was increasingly utilized in countries such as Egypt, constituting 40% of global HCV treatment that year [2]. By 2023, in more than 100 LMICs, over 87% had registered at least one DAA therapy for treatment [3].

The efficacy of DAAs lies in their targeting of multiple stages of the viral life cycle, delaying viral replication and reducing viral load [4]. DAAs accomplish this by the selective attack and inhibition of several HCV proteins, namely NS5A, NS5B polymerase, and nonstructural protein 3/4A protease [1]. These mechanisms lead to improvements in sustained virological response (SVR) and reduce the overall health impacts associated with HCV [4]. However, despite such advances, significant challenges remain in the way of the global eradication of the disease. Over 70 million individuals affected by HCV have not received treatment [2], suggesting problems with access and/or delivery.

HCV prevalence is disproportionately higher in LMICs that have lower Gross National Income (GNI) per capita and often pale in comparison to other countries regarding healthcare delivery, infrastructure, and other factors that contribute to poorer health outcomes [5]. According to the World Health Organization (WHO), two-thirds of the global HCV burden can be attributed to just fifteen countries [2,6]. Eight of these countries—India, Pakistan, Nigeria, Ukraine, Uzbekistan, Vietnam, Bangladesh, and Ethiopia—are classified as LMICs by the World Bank [5,6]. Individuals in these regions face significant barriers to receiving DAAs, and a substantial proportion of those infected remain unaware of their condition [7,8].

To work toward achieving the WHO’s goal of eliminating HCV as a public health threat by 2030, it is crucial to understand and address the major challenges that LMICs face in providing effective HCV treatment. This paper aims to explore the common obstacles encountered across LMICs in their efforts to combat HCV as well as identify potential interventions that have exhibited success in addressing and mitigating these barriers.

## 2. Global Overview

The incidence of hepatitis and liver diseases is significantly lower in developed countries like the United States compared to LMICs such as India and Pakistan [9]. Per 2023 data, Pakistan has one of the highest incidences of HCV globally (9,311,083 infections), followed by India (6,032,021 infections). India has more than double the HCV incidence of other LMICSs such as Nigeria [9]. The United States reported 2,445,824 HCV infections in 2023, with other LMICs including Ukraine, Uzbekistan, Bangladesh, Vietnam, and Ethiopia also contributing substantially to the global burden [9]. Factors such as inadequate testing and social stigma likely contribute to these disparities.

With respect to HCV diagnosis rates, countries such as Bangladesh, Pakistan, and India lag significantly behind the United States, with the highest diagnosis rate at approximately 30% (Figure 1 [9]). Treatment rates, however, remain low across the board. In Pakistan and India, only 1–2% of diagnosed individuals receive treatment, compared to about 6% in the United States. Several LMICs, including Bangladesh, Ethiopia, Nigeria, Vietnam, and Ukraine, report treatment rates of less than 1%. Uzbekistan is characterized by both diagnosis and treatment rates below 1% (Figure 1 [9]).

In terms of disease burden, Disability-Adjusted Life Year (DALY) rates highlight the impact of limited resources and restricted access to antiviral treatment in LMICs. In 2021, the DALY rate for chronic Hepatitis C in the United States was 202.34 years lost per 100,000 population (Figure 2 [10]). In comparison, LMICs such as Pakistan, Uzbekistan, and Ukraine reported significantly higher DALY rates, losing 407.61, 341.48, and 461.69 years, respectively (Figure 2 [10]).

When examining the current global picture of HCV, it is important to consider factors beyond incidence and treatment rates. Transmission, while largely uniform in spreading via contact with infected blood, can vary in specific modes of disease transfer by country. Preceding widespread testing for the virus, HCV was primarily transferred via inadequate medical hygiene in healthcare settings, through interventions such as blood transfusions and dialysis [11]. Through the decades, this mode of transmission has diminished in high-income countries, but remains an issue in LMICs; in countries like Pakistan, “unsafe medical injections” contribute to more than 40% of new HCV infections [11]. Engagement in risky behaviors, including having multiple sexual partners, receiving tattoos and piercings, and injection drug use (IDU) all increase the risk of developing HCV [11]. IDU especially has risen to become a major problem in Western countries like the U.S. with the concurrent rise in opioid use [11]. Rates of IDU-related HCV infections have also risen among LMICs, most notably in select South American and Asian nations [11].

Despite global efforts to achieve the WHO’s goal of eliminating HCV as a public health threat by 2030, many LMICs continue to face significant barriers to effective treatment. These include limited access to healthcare resources, inadequate diagnostic facilities, and high treatment costs, rendering disease burden reduction in these regions particularly challenging.

## 3. Materials and Methods

Information was drawn from publicly accessible information online from health organization databases including WHO, the Polaris Observatory, and the Institute for Health Metrics and Evaluation (IHME) using Google search tools. All three of these respective databases contained current and comprehensive statistical data about global HCV prevalence, diagnosis and treatment rates, and specific measures such as DALYs. Search terms included “hepatitis C”, “HCV”, and “LMICs” in combination with “direct-acting antivirals”, “treatment”, “treatment barriers”, “treatment obstacles”, “financial barriers”, “knowledge”, and “awareness”. An informal literature review was conducted using PubMed and Google Scholar with relevant MeSH terms “hepatitis C, chronic”, “developing countries”, “low-income countries”, and “lower middle-income countries” to investigate barriers to HCV treatment in LMICs.

Inclusion criteria included peer-reviewed papers in the English language, published between January 2011 and January 2024. For treatment barriers, studies that either broadly discussed LMIC obstacles or explored LMIC-specific challenges (such as for Pakistan, Egypt, or the Democratic Republic of the Congo) were included in the review. Pertaining to recommendations and solutions, studies that discussed results from the implementation of initiatives that addressed restrictions to HCV were included, with research conducted both in LMICs and high-income countries like the U.S. and Canada being incorporated. Citations that were not peer-reviewed, were not in English, were outside the publication period, did not pertain to the research topic, or only detailed challenges in high-income countries were excluded from the review. From the thirty-four references collected, reported HCV treatment barriers and proposed solutions were abstracted and compiled according to theme. Shared barriers among the LMICs were identified as overarching themes, with separate subgroups categorized as “lack of knowledge”, “social stigma”, and “financial barriers”. These commonalities across LMICs were recognized and synthesized as recommendations. A similar method was employed to propose solutions to these barriers, and upon review, distinct themes of “healthcare interventions”, “community interventions”, and “strategic formation of international relationships” emerged.

## 4. Results

### 4.1. Barriers in Low- and Middle-Income Countries (LMICs)

#### 4.1.1. Lack of Knowledge and Social Stigma

In LMICs, healthcare workers (HCWs) often lack sufficient knowledge and awareness about diseases such as HCV. For example, in the Democratic Republic of the Congo, midwives receiving antenatal education were instructed on nutrition and diseases such as HIV and malaria but received no training on the Hepatitis B Virus (HBV) or HCV [12]. Additionally, HCWs were often unaware that HCV is a notifiable disease, leading to under-reporting [13]. In Nigeria, only 12% of physicians were familiar with the disease notification system, and just 24% were able to fill out the necessary notification forms and knew where to obtain them [13]. These knowledge gaps, combined with the existing shortage of HCWs and medical supplies, contribute to low rates of HCV diagnosis and treatment in these regions.

In LMICs, the lack of knowledge and understanding of the disease among the general population also represents an obstacle. Across sub-Saharan Africa, many people were unaware of key aspects of HCV, such as transmission methods, risk factors, and prevention strategies [12]. In Egypt, a country that had one of the highest HCV prevalence rates globally, studies have shown that even among those diagnosed with HCV, the level of understanding was very low. Misconceptions about transmission are common; for instance, in a study assessing HCV knowledge, some believed that HCV can be spread through shaking hands (10.2%) or working closely with an infected person (14.5%). Only about one-third of participants were aware of the risks of vertical transmission from mother to child (31.3%) [14]. Notably, 80.8% of patients with HCV in Egypt first learned of their condition only after a routine health visit [14]. In a study from Pakistan involving a group composed predominantly of healthcare professionals, only 22% of study participants demonstrated comprehensive knowledge of HCV [15]. In high-risk groups, such as people who inject drugs (PWIDs) in India, over half of the participants in one study were unaware of HCV as a disease, let alone their increased susceptibility due to drug injection [16].

Misinformation shrouding HCV in LMICs has led to stigmas that have proven harmful to at-risk populations such as PWIDs. In Kyrgyzstan, PWIDs with HCV were surveyed on knowledge regarding the disease and obstacles faced. Among the subjects surveyed, multiple participants reported “fear of stigma and discrimination from healthcare workers” as one of the main reasons why many PWIDs with HCV are reluctant to receive treatment [17]. In Kenya, the stigma surrounding HCV has led to a reluctance to share HCV status with close family and friends due to the risk of isolation and damage to relationships [18]. Beyond social ostracization, study participants reported difficulties keeping antiviral medication at home for fear of discovery by family members [18]. These findings encapsulate the kinds of societal views that have affected testing rates and stymied the treatment of HCV in LMICs [16].

#### 4.1.2. Financial Barriers

Since 2015, the introduction of generic manufacturers into the HCV antiviral market has significantly reduced the cost of direct-acting antivirals (DAAs). For instance, the price of the generic version of sofosbuvir has decreased by over 70% in LMICs, pricing at USD 48 in Egypt and USD 25 in Pakistan for a twelve-week treatment course [1,19]. Beyond DAAs, LMICs have also seen improvements in access to affordable HCV screening, with immunoassays priced between USD 10 and USD 25, making these tests more widely accessible [20].

Despite the increased affordability of DAA treatment on a per-unit basis, in many LMICs, funding-related obstacles at the administrative level continue to hinder the institutional implementation of HCV treatment. The challenge lies further along the care cascade, where after serological testing, patients must undergo confirmatory tests such as nucleic acid testing (NAT).

LMICs often lack the necessary infrastructure to support these diagnostics as healthcare systems struggle with the prompt processing of whole blood, the cold storage of samples, and the high cost of instruments, reagents, and trained technicians needed for these assays [20]. Inadequate international financing for confirmatory testing has impeded laboratories, such as those run by Biocentric and Roche, from incorporating diagnostic HCV tests, resulting in reduced testing throughput [21]. Moreover, limited funding affects the accessibility of services to all population groups. For example, in India and Indonesia, where government-facilitated HCV testing and treatment are available, limited funding restricts screening primarily to blood donors [21]. Other high-risk groups, including PWIDs, pregnant women, children born to mothers infected with HCV, individuals with HIV and other chronic illnesses, and prisoners, received screening in fewer than half of the LMICs surveyed (Table 1 [21]).

A study assessing HCV and HBV testing practices across various LMICs revealed that complimentary HCV RNA testing was only reported in 5–30% of the countries surveyed, including Georgia, Indonesia, South Africa, Malaysia, and Turkey [21]. Furthermore, advanced screenings for late-stage symptoms of HCV, such as liver disease and fibrosis, were conducted in only two countries: Georgia and Macedonia [21]. Low testing practices can be contextualized by the fact that over 45% of the LMICs reporting funding for testing note that the funding is not government-sponsored but derived from patients through self-payments or private insurance [21].

The absence of sufficient funding and adequate infrastructure to support HCV testing and treatment in LMICs can often be attributed to poor governance and corruption. Research conducted in South and Southeast Asian LMICs has revealed how health delivery outcomes suffer from harmful governmental practices, including bribery, restrictive enrollment into health insurance, and superfluous payments [22]. In Bangladesh, one-third of patients receiving care at district hospitals had to make extra payments for hospital admission [22]. Such practices not only place an increased financial burden on lower-income families but also decrease public trust in key social institutions, healthcare, and governmental services [22]. International healthcare providers treating HCV in both the Middle East and Africa cited such increased healthcare expenses and poor insurance coverage as major obstacles in the delivery of HCV care [23].

## 5. Discussion and Recommendations

Addressing the barriers to HCV treatment and care requires a multifaceted and comprehensive approach, as the complexity of the issue precludes a single, overarching solution. Challenges exist across various aspects of the care continuum, as well as within the broader healthcare infrastructure. Therefore, solutions must be tailored to the specific needs of each country or healthcare facility to optimize outcomes.

### 5.1. Healthcare Interventions

#### 5.1.1. Increasing Awareness

A key barrier to effective HCV treatment is the lack of knowledge and awareness among healthcare providers regarding HCV management. Educational reform targeting HCWs is essential to bridge this knowledge gap. Providers must be well informed about HCV transmission, immunization, risk management, infection control, treatment options, and payment plans to improve patient care. Studies on HCW compliance with infection prevention control (IPC) measures for various infectious diseases, including influenza and HCV, have shown that greater adherence is observed among HCWs actively involved in IPC committees, those who stay informed through medical journals, and those with access to IPC guidelines [21]. Conversely, poor adherence to IPC protocols has been reported among HCWs with inaccurate information about vaccine efficacy and safety, as well as those lacking the necessary equipment to implement safety measures [24]. Implementing educational reforms to enhance IPC compliance and reduce healthcare-acquired infections (HAIs) has proven effective in dispelling vaccine misconceptions, increasing the use of isolation measures, and reducing common injuries such as needlestick incidents [24].

#### 5.1.2. Telementorship

HCV, as an infectious disease, requires reform initiatives to achieve significant improvements in health outcomes and reduce global prevalence. The implementation of telementorship programs, for example, has shown promise in enhancing HCV care globally. Telementorship refers to regular, monthly online meetings of HCWs to provide opportunities for continued education on HCV, addressing frequently asked questions about treatment initiation and termination, HCV management during pregnancy, and treatment in patients with comorbidities [25]. Telementorship facilitates communication, enhances provider knowledge, and promotes team-based learning and problem-solving. This approach has improved HCV care delivery, particularly in underdeveloped and rural regions [25]. The flexibility of telementorship is effective not only in urban settings but also in countries with varying economic status, including LMICs such as India, Egypt, and Pakistan, supporting its potential for global application [25]. Initial evaluations of sites utilizing telementorship showed comparable outcomes in treatment factors such as sustained virological response, viral suppression, and treatment initiation as compared to sites not using this approach [25]. Over time, however, HCWs participating in telementorship reported increased self-efficacy and job satisfaction, and patients benefited from more timely and effective HCV treatment [25]. The ability to connect HCWs across rural and urban regions fosters knowledge exchange and collaboration in infectious disease care, making telementorship an important area for more investment. With further development, telementorship has the potential to significantly reduce knowledge gaps among providers and improve treatment outcomes for HCV patients globally.

### 5.2. Community-Based Interventions

#### 5.2.1. Patient Education and Awareness

Many patients in both LMICs and non-LMICs alike are unaware of their infection status, modes of transmission, treatment options, health insurance coverage, and preventative measures. These factors contribute to the persistence of HCV-related stigma, underdiagnosis, reduced treatment initiation, and lower treatment completion rates. To improve health outcomes, it is crucial that at-risk populations and those already diagnosed with HCV receive targeted educational interventions.

A notable example of the impact of such interventions can be seen among the unhoused population in the United States, where community-based educational programs have significantly increased HCV awareness. These programs, which often include HCV screenings, have been shown to enhance the likelihood of individuals engaging in treatment [26]. Educational sessions typically involve the delivery of informational PowerPoint presentations by HCWs to the at-risk population at various locations [26]. Before these educational interventions, 70.68% of study participants in one study were inclined to agree to treatment, but after receiving HCV education, this number increased to 86% [26]. This demonstrates the potential of patient education initiatives to improve treatment uptake and overall health outcomes among vulnerable populations. While these findings are particularly relevant to the U.S., similar community-based interventions could be equally effective and cost-efficient in LMICs.

Elsewhere, the use of similar presentations and multimedia aids—such as videos, PowerPoint presentations, and pamphlets—that were tailored to local languages and cultural contexts have proven successful in raising HCV awareness [27]. A systematic review of HCV educational interventions in India, the United Kingdom, and Pakistan, involving nearly 70,000 participants, found that 14 of the 15 interventions were successful in promoting testing and education [27]. Integrating these educational efforts with HCV health screenings and public health services provides a comprehensive approach to bridging the HCV knowledge gap, increasing access to testing, and subsequently boosting treatment initiation rates [27].

The key takeaway from existing research, it seems, is that educational interventions for HCV need not be overly complex or technical in their delivery. However, they must include relevant and accurate information on HCV epidemiology, transmission, and treatment to effectively serve as harm reduction measures and educational resources for patients. While didactic classes and lessons are effective in educating HCWs, community-based approaches are ideal for reaching the public and vulnerable populations.

#### 5.2.2. Physical Barriers

When it comes to receiving HCV treatment in LMICs, transportation to hospitals and clinics remains difficult for many. Several countries have utilized both food and transportation vouchers as an incentive for at-risk patients to seek treatment. With programs designed to foster HIV care in Kenya, for example, the implementation of food vouchers led to a significant increase in voluntary compliance among men over a two-month period. Travel vouchers were also provided, with monetary value estimated to be two to three days’ worth of income [28]. In Uganda, transportation vouchers were reported as being a significant factor in HIV prevention, enabling men to overcome travel barriers and undergo circumcision procedures [29].

Meanwhile, countries such as the United States and Canada have taken a different approach by delivering services *to* at-risk individuals for HCV, seeking to eliminate any travel restrictions. In Vancouver, Canada, community pop-up clinics were met with success at targeting high-risk populations, specifically PWIDs, where antibody testing and consultation were provided on-site in various locations around the downtown region, and meal vouchers were also offered to individuals who scheduled follow-up appointments [28]. Of patients who tested positive for HCV, 76% would go on to meet with an infectious disease physician regarding their disease status and 50% of patients would visit a clinic to further their care [30]. This study also highlighted the success of vouchers, as 100% of patients redeemed their meal vouchers upon clinic visitation [30].

Similarly, in the United States, a mobile health clinic also demonstrated the value of providing HCV screening and treatment directly to vulnerable populations. A considerable percentage of the patients seen in clinics who underwent testing were often uninsured (42.8%), and many individuals were reported to be unsuspecting of their disease-positive status [30]. There was also a notable increase in patient initiation of treatment (49.6%) in comparison to the national average of HCV treatment initiation (9.9–15.5%). PWIDs and uninsured people made up a sizeable proportion of individuals who were HCV-positive and initiated treatment, at 85.5% and 74.5%, respectively [31]. In addition, 86% of patients would go on to complete their HCV treatment [31]. The success of voucher programs and the ability of community and mobile clinics to reach and treat high-risk and underserved populations thus represent two viable avenues for increasing HCV care and otherwise overcoming related barriers in LMICs.

### 5.3. Strategic Investments and International Support

#### 5.3.1. Enhancing Financial Investment

While educational interventions are crucial for mitigating HCV, a broader and more impactful solution, particularly for LMICs, is to increase financial investment in healthcare infrastructure. Increased funding could address a range of issues, from enhancing screening and diagnostic capabilities to improving treatment initiation and adherence rates. For example, new diagnostic tools such as reflex HCV RNA tests, which eliminate the need for patients to undergo a second blood test, could significantly reduce the number of missed chronic infections [32]. Additionally, implementing rapid point-of-care (POC) antibody testing in rural and underserved areas could allow for immediate identification and diagnosis of HCV infections [32]. The success of these new diagnostic modalities in LMICs would depend on increased financial investment specifically allocated to HCV care (Table 2 [32]).

#### 5.3.2. Leveraging International Aid and Partnerships

Realistically, funding HCV care and treatment in LMICs is challenging without international support from global organizations. Unlike other infectious diseases such as HIV and malaria, viral hepatitis has historically received less global funding from associations like the Global Fund and PEPFAR [33]. However, financial support can be highly effective, as demonstrated by successful implementation of HCV elimination programs in Georgia through a combination of national funds and international investments from the CDC and Unitaid [33]. Even when direct funding is unattainable, partnerships with global agencies can significantly assist in HCV elimination efforts. Another such example is the Clinton Health Access Initiative (CHAI), which has successfully partnered with several LMIC governments to improve access to HCV treatment and reduce the prices of DAAs [34]. These efforts have led to significant improvements in diagnostic testing prices, with rapid diagnostic tests costing less than USD 1.00 per test in several CHAI-assisted countries. Testing volumes have also increased, with Rwanda conducting upwards of 1.5 million tests, and the overall treatment success rate among individuals infected with HCV reaching 90% across LMICs such as Rwanda, Cambodia, Vietnam, India, and Nigeria [34].

While there are numerous contributing factors towards the successful implementation of HCV elimination programs in the CHAI countries, a commonality is that they are all receiving global support from major organizations such as the WHO, FIND, and MSF. The WHO has partnerships with several LMICs including India, Vietnam, Nigeria, Cambodia, and Myanmar [34]. MSF and FIND have also formed ties with India and Myanmar [34]. Aside from funding, these groups offer a uniform framework and guidelines for HCV elimination built around the decentralization of HCV care, expansion of HCV services [34], and streamlining of care provision.

## 6. Limitations

Methodologically, a broad net was cast to identify relevant research and data with respect to HCV treatment and interventions in LMICs. The exclusion of non-English sources may preclude a truly comprehensive picture and in the context of LMICs, the potential loss of cultural context should be noted with the use of only English-based sources. Nevertheless, given the scant non-English literature pertaining to the topic, the overall potential impact of this selection bias is deemed to be low.

While there is more recent literature on solutions and interventions in Western countries like the U.S. and Canada, studies on HCV in the LMIC context are rarer and challenging to identify. Accordingly, these findings may not account for unpublished current trends in HCV data, treatment, and solutions.

A myriad of gaps remains to be addressed regarding HCV research in LMICs. More research on HCV treatment should be conducted in LMICs where the bulk of HCV infections are to ascertain important contextual cultural factors and overall treatment efficacy. In addition, the demonstration(s) of successful community-level interventions for HCV treatment have largely only been undertaken in Western countries. A logical next step would be to assess these types of interventions in LMICs to determine whether similar successes can be achieved.

## 7. Conclusions

Achieving global HCV elimination remains a significant challenge, particularly in LMICs where barriers exist at all levels from healthcare infrastructure to public awareness. Major investments are needed to expand HCV testing and screening capacity in these nations, as well as greater educational efforts to address the lack of knowledge that stymies treatment initiation and adherence. Prioritizing educational interventions for both healthcare providers and the public is essential for overcoming these challenges, and great promise has been shown in programs that bring HCV education and testing *to* affected populations. While global progress has been made, particularly with the introduction of DAAs, sustained investment and coordinated efforts are vital if the WHO’s HCV elimination target is to be achieved.

## Figures and Tables

**Figure 1 healthcare-13-00043-f001:**
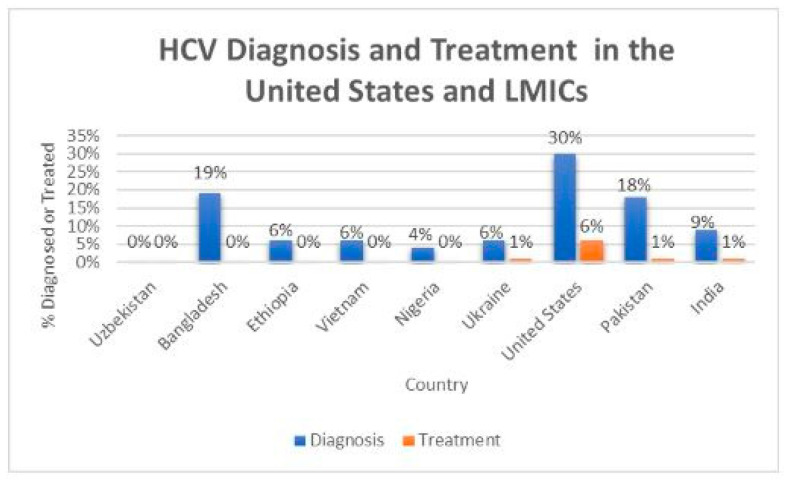
Comparative Hepatitis C Virus (HCV) diagnosis and treatment rates.

**Figure 2 healthcare-13-00043-f002:**
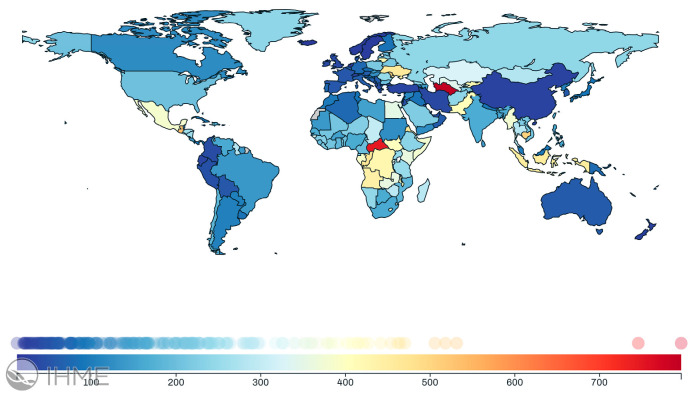
Global distribution of Disability-Adjusted Life Year (DALY) rates caused by chronic Hepatitis C Virus (HCV). Data from the Institute for Health Metrics and Evaluation (IHME) global health data (GBD) tool at https://vizhub.healthdata.org/gbd-compare/ (accessed on 22 December 2024).

**Table 1 healthcare-13-00043-t001:** HCV testing rates by target population in LMICs.

Target Population	HCV Testing (RDT, EIA, RIA)
Blood Donors	86.40%
People Who Inject Drugs	50%
Pregnant Women	31.80%
Children Born to Mothers Infected with HCV/HBV	54.50%
Chronically Ill	31.80%
People Living with HIV	40.90%
Prisoners	22.70%

RDT/EIA/RIA—rapid diagnostic test/enzyme immunoassay/radioimmunoassay. Data from “Values, preferences, and current hepatitis B and C testing practices in low- and middle-income countries: results of a survey of end users and implementers” [21].

**Table 2 healthcare-13-00043-t002:** Barriers, challenges, and potential solutions for HCV care in LMICs.

Barriers	Challenges	Solutions
Knowledge	HCW awareness about HCV care	HCW educational reform
	Knowledge about HCV transmission and treatment	Telementorship
	Social stigma	Multimedia community education
Personal Costs	Cost of travel to healthcare centers	Meal and travel vouchers
	Time to travel to healthcare centers	Pop-up community clinics
		Mobile health clinics
Healthcare Infrastructure	Inadequate facilities to implement nucleic acid testing (NAT), cold storage, confirmatory testing	Increased governmental investment into healthcare
	Inability to test high-risk populations	Ally with international aid organizations

## Data Availability

The original data presented in this study are openly available in the Global Health Data Exchange at ghdx.healthdata.org.

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
