# Peer review of "Understanding Barriers to Hepatitis C Antiviral Treatment in Low–Middle-Income Countries"

_healthcare, 2024, doi:10.3390/healthcare13010043_

Round 1
Reviewer 1 Report
Comments and Suggestions for Authors
This paper is timely and interesting! I suggest that you state some possible limitations of the study. Also, the conclusion is relatively short. I think that adding some more about future directions of research, along with limitations of this study and other studies would be helpful.
I suggest the authors add more information about what direct-acting antiviral agents (DAAs) and Hepatitis 12 C Virus (HCV) are. Also, info about how HCV is transmitted in various countries (by income level) would be important. Also, it would be helpful for the authors to address gender, rural/urban, education level, and SES differences in access to testing and treatment across the countries. The conclusion mentions some of the efforts and strategies that could be put into place to help decrease barriers to access to info and treatment, but I think that the authors should be more specific about how these interventions and strategies would look and work, especially in very low-income nations. For instance, they state in line 337, "Prioritizing educational interventions for both healthcare providers and the public is essential for overcoming these challenges," but I think more information about what that educational interventions would look like.
Author Response
Reviewer 1
Thank you for your comments and suggestions. They have been addressed below.
- I suggest that you state some possible limitations of the study. Also, the conclusion is relatively short. I think that adding some more about future directions of research, along with limitations of this study and other studies would be helpful.
Thank you for your comments and suggestions. As suggested, a section that discusses the limitations of the study as well as future directions of research was added under Limitations, beginning at line 408.
- I suggest the authors add more information about what direct-acting antiviral agents (DAAs) and Hepatitis 12 C Virus (HCV) are.
Additional context regarding what the HCV virus is has been provided in the introduction (line 69-72) as well as DAAs (line 87-89).
- Also, info about how HCV is transmitted in various countries (by income level) would be important.
Under the global overview section, beginning at line 129, more in depth discussion over the transmission of the virus and how it differs among the high and low-middle income countries, with high-risk behaviors, hospital transmission, and injection drug use discussed.
- Also, it would be helpful for the authors to address gender, rural/urban, education level, and SES differences in access to testing and treatment across the countries.
Our findings did not uncover sufficiently detailed information to address gender, rural/urban, educational level and SES differences in access to testing and treatment across the countries. We concur that these would be interesting topics to explore for further study. The ways in which interventions and strategies might look like in LMICs are addressed in detail in the discussion and recommendations.
- The conclusion mentions some of the efforts and strategies that could be put into place to help decrease barriers to access to info and treatment, but I think that the authors should be more specific about how these interventions and strategies would look and work, especially in very low-income nations. For instance, they state in line 337, "Prioritizing educational interventions for both healthcare providers and the public is essential for overcoming these challenges," but I think more information about what those educational interventions would look like.
We agree that specifics of how interventions would work in LMICS should be discussed. In the manuscript, we make effort to do this throughout the discussion section, with specific examples being seen in lines 298-302, 330-335, 345-350, 393-398.
Reviewer 2 Report
Comments and Suggestions for Authors
Thank you for submitting your manuscript, "Understanding Barriers to Hepatitis C Antiviral Treatment in Low-Middle Income Countries." Your work highlights a critical and timely issue in global health, addressing the multifaceted challenges LMICs face in achieving equitable access to Hepatitis C treatment. Your comprehensive approach, combining an in-depth literature review with actionable recommendations, demonstrates a strong commitment to advancing the WHO’s goal of eliminating HCV as a public health threat by 2030. We appreciate the thoroughness of your analysis and the innovative solutions you propose, such as telementorship and community-based interventions. Your manuscript provides valuable insights and serves as a meaningful contribution to the field. Thank you for your efforts in shedding light on this important topic.
Strengths of the manuscript
· The manuscript addresses a critical global health issue by exploring barriers to Hepatitis C treatment in LMICs, aligning with the WHO's goal to eliminate HCV as a public health threat by 2030.
· The introduction and background sections provide a comprehensive overview of the problem, detailed descriptions of HCV treatment challenges, and relevant statistics.
· The manuscript is logically structured, moving systematically through barriers, interventions, and recommendations.
· The use of Disability-Adjusted Life Years (DALY) and comparisons between LMICs and developed countries adds valuable quantitative depth.
· Including telementorship and community-based interventions demonstrates an effort to propose actionable, context-sensitive solutions.
· The manuscript draws on diverse sources, including WHO reports, systematic reviews, and case studies, ensuring a robust evidence base.
Suggested areas for improvement:
- The study's objectives could be explicitly stated in the introduction to help readers understand the focus and intended contributions of the research.
- The "Materials and Methods" section lacks detailed explanations of how literature was reviewed and selected. Clarifying inclusion/exclusion criteria, database selection, and the thematic analysis process would improve transparency.
- While the manuscript discusses LMICs broadly, it could benefit from a more in-depth discussion of how cultural and regional factors influence HCV treatment barriers and solutions.
- Including a figure on HCV diagnosis and treatment rates is helpful, but more visual aids (e.g., maps showing DALYs or treatment coverage) could enhance comprehension and engagement.
- Table 1 could be expanded to include more granular data on specific target populations and interventions.
- Although barriers and solutions are identified, the discussion could explore deeper into why specific strategies succeed or fail in LMICs, incorporating lessons learned from other health interventions like HIV or TB.
- The recommendations section could benefit from a clearer roadmap or framework outlining specific steps for LMICs and international organizations to take.
- Some sentences are lengthy and could be simplified for better readability. For instance, the paragraph discussing DALYs could be streamlined to emphasize the key findings.
Please consider:
· In the abstract, a concise summary of the results and key recommendations would improve its impact.
- In the introduction, adding a global comparison of treatment accessibility could provide additional context.
- While financial and knowledge barriers are discussed in the barriers section, addressing systemic issues like corruption or inadequate governance would provide a more holistic view.
- Citations are thorough but slightly dated in some sections. Incorporating more recent studies (2023–2024) could improve relevance.
Suggested Additions:
- Including brief case studies from LMICs that have made progress in HCV treatment could illustrate practical applications of the recommendations.
- Adding a section on how to measure the success of proposed interventions would be valuable.
- Highlighting gaps in current knowledge and areas for future study would position the manuscript as a foundation for ongoing research.
The manuscript contributes significantly to understanding barriers to HCV treatment in LMICs and offers innovative solutions. Addressing the identified areas for improvement could enhance the findings' clarity, applicability, and impact.
Author Response
Reviewer 2
We certainly appreciate this feedback and look forward to sharing our work with the journal’s readership.
Improvement:
- The study's objectives could be explicitly stated in the introduction to help readers understand the focus and intended contributions of the research.
As suggested, the objectives of the paper are now more clearly stated at the end of the introduction at lines 105-107.
- The "Materials and Methods" section lacks detailed explanations of how literature was reviewed and selected. Clarifying inclusion/exclusion criteria, database selection, and the thematic analysis process would improve transparency.
To improve transparency, more detail is provided regarding the process of the review, including why databases were chosen, the inclusion and exclusion criteria, as well as how the themes/groupings of the recommendations was determined. This is found in lines 158-160, 168-177, 178-184.
- While the manuscript discusses LMICs broadly, it could benefit from a more in-depth discussion of how cultural and regional factors influence HCV treatment barriers and solutions.
Our findings did not uncover sufficiently detailed information to independently address
gender, rural/urban, educational level and SES differences in access to testing and
treatment across the countries. We concur that these would be interesting topics to further
explore.
- Including a figure on HCV diagnosis and treatment rates is helpful, but more visual aids (e.g., maps showing DALYs or treatment coverage) could enhance comprehension and engagement.
Beginning at line 132, another visual aid was added depicting 2021 global DALY rates to help represent the discussion in the previous paragraph.
- Table 1 could be expanded to include more granular data on specific target populations and interventions.
No additional data on specific target populations and interventions were identified from
our sources for inclusion.
- Although barriers and solutions are identified, the discussion could explore deeper into why specific strategies succeed or fail in LMICs, incorporating lessons learned from other health interventions like HIV or TB.
We make an attempt to justify how implementation of these interventions could improve in LMICS. Examples are in lines 300-302 regarding telehealth and the research that indicates it will positively improve factors such as treatment initiation, and SVR. Lines 330-335 discuss successful implementation of educational interventions in LMICs like Pakistan. Lines 345-350 discuss HIV interventions to attenuate physical barriers of transportation by incorporation of travel vouchers and propose its implementation in the context of HCV treatment in countries like Uganda and Kenya.
- The recommendations section could benefit from a clearer roadmap or framework outlining specific steps for LMICs and international organizations to take.
While providing a roadmap/ framework for LMICs and international organizations could certainly serve as a helpful resource, it is beyond the scope of this particular manuscript. Consideration of a future paper that includes a deep dive into contextual factors for each LMIC and organization informing such a roadmap is in the works.
- Some sentences are lengthy and could be simplified for better readability. For instance, the paragraph discussing DALYs could be streamlined to emphasize the key findings.
As recommended, we reviewed the paper in efforts to streamline and reduce length. The paragraph discussing DALYs currently has the key findings from Figure 2 in which countries like Pakistan and Ukraine have higher DALYs that other nations.
Please consider:
- In the abstract, a concise summary of the results and key recommendations would improve its impact.
The abstract was edited to be more concise and clearer, explicitly stating the results and the review’s recommendations, from line 55-62.
- In the introduction, adding a global comparison of treatment accessibility could provide additional context.
More context is added to describe the evolution of treatment accessibility globally since the introduction and approval of DAAs, from lines 75-85.
- While financial and knowledge barriers are discussed in the barriers section, addressing systemic issues like corruption or inadequate governance would provide a more holistic view.
We recognize the importance of these issues and now touch upon it in the discussion from lines 256-265, but note that no recent data or articles were found that specifically addressed this.
- Citations are thorough but slightly dated in some sections. Incorporating more recent studies (2023–2024) could improve relevance.
Search was conducted through 2024.
Suggested Additions
- Including brief case studies from LMICs that have made progress in HCV treatment could illustrate practical applications of the recommendations.
Results of interventions in LMICs are directly incorporated and discussed in the
paper.
- Adding a section on how to measure the success of proposed interventions would be valuable.
Measurement of success of proposed interventions along with the roadmap as
suggested will be the focus of a future paper.
- Highlighting gaps in current knowledge and areas for future study would position the
manuscript as a foundation for ongoing research.
Beginning at line 425, future directions of research is discussed including more research done in LMICs to improve data and add cultural context, as well as conduct studies that would implement HCV interventions in the LMICs.
Reviewer 3 Report
Comments and Suggestions for Authors
The paper aims to understand barriers to hepatitis c antiviral treatment in low-middle income countries. I commend the authors on the meticulousness and rigor with which you conducted this review.
My questions are as follows:
1) On Page 1, line 38: “Federal Drug Administration”. Do you mean the “USA food and drug administration”? Based on this sentence, it seems that DAAs were approved in 2011 for any country. I want to double check with the authors if I understand this sentence correctly. If not, it would be important to mention different timelines for the first DAA approval in selected countries or to add a supplemental file listing the timeline for the first DAA approved in different selected low-middle income countries in the introduction part.
2) On Page 3, line 100: I understand the strategy of excluding literature not published in English. However, there may be publication bias resulting from not including papers in other languages, especially when focusing on low-middle income countries. It seems that the authors did not include a limitation part in the manuscript. If the formatting aligns with the requirements from the journal, please ignore this comment. Otherwise, I think it would be great to list the limitations of this review in the discussion part.
3) On page 4, lines 177-181: I was wondering whether these are the footnote for table 1?
I look forward to learning from your responses.
Author Response
Reviewer 3
Thank you for your comments and suggestions. They have been addressed below.
- On Page 1, line 38: “Federal Drug Administration”. Do you mean the “USA food and drug administration”? Based on this sentence, it seems that DAAs were approved in 2011 for any country. I want to double check with the authors if I understand this sentence correctly. If not, it would be important to mention different timelines for the first DAA approval in selected countries or to add a supplemental file listing the timeline for the first DAA approved in different selected low-middle income countries in the introduction part.
Yes, you are correct that the “Federal Drug Administration” refers to US food and drug administration. This was subsequently clarified in the manuscript. Upon further look, 2011 approval of DAA use refers to the U.S specifically. A brief timeline of increased use and development of the DAAs was discussed in the introduction, including Egypt, Mongolia and Pakistan
- On Page 3, line 100: I understand the strategy of excluding literature not published in English. However, there may be publication bias resulting from not including papers in other languages, especially when focusing on low-middle income countries. It seems that the authors did not include a limitation part in the manuscript. If the formatting aligns with the requirements from the journal, please ignore this comment. Otherwise, I think it would be great to list the limitations of this review in the discussion part.
Thank you for these suggestions. The issue of selection bias, as well as other limitations were addressed in a newly formed limitations section in the manuscript starting on line 408, following part 3.2, It discusses selection bias of excluding non-English papers as well as the limited amount of current data available regarding the research topic.
- On page 4, lines 177-181: I was wondering whether these are the footnote for table 1?
I look forward to learning from your responses.
Yes, you are correct. That is the corresponding footnote that will follow Table 1.